# Genomic Profiling of Highly Aggressive Musculoskeletal Sarcomas Identifies Potential Therapeutic Targets: A Single-Center Experience

**DOI:** 10.3390/cancers18010139

**Published:** 2025-12-31

**Authors:** Alessandro Parra, Emanuela Palmerini, Maria Antonella Laginestra, Cristina Ferrari, Stefania Cocchi, Elisa Simonetti, Evelin Pellegrini, Alessandra De Feo, Giovanna Magagnoli, Giorgio Frega, Davide Maria Donati, Marco Gambarotti, Toni Ibrahim, Katia Scotlandi, Lorena Landuzzi, Laura Pazzaglia

**Affiliations:** 1Department of Pathology, IRCCS Istituto Ortopedico Rizzoli, 40136 Bologna, Italy; alessandro.parra@ior.it (A.P.); stefania.cocchi@ior.it (S.C.); giovanna.magagnoli@ior.it (G.M.); marco.gambarotti@ior.it (M.G.); 2Osteoncology, Bone and Soft Tissue Sarcomas, and Innovative Therapies Unit, IRCCS Istituto Ortopedico Rizzoli, 40136 Bologna, Italy; emanuela.palmerini@ior.it (E.P.); giorgio.frega@ior.it (G.F.); toni.ibrahim@ior.it (T.I.); 3Laboratory of Oncology Research and Functional Genomics, IRCCS Istituto Ortopedico Rizzoli, 40136 Bologna, Italy; mariaantonella.laginestra@ior.it (M.A.L.); cristina.ferrari@ior.it (C.F.); elisa.simonetti@ior.it (E.S.); evelin.pellegrini@ior.it (E.P.); alessandra.defeo@ior.it (A.D.F.); katia.scotlandi@ior.it (K.S.); laura.pazzaglia@ior.it (L.P.); 4Clinica Ortopedica e Traumatologica III a Prevalente Indirizzo Oncologico, IRCCS Istituto Ortopedico Rizzoli, 40136 Bologna, Italy; davidemaria.donati@ior.it

**Keywords:** sarcomas, osteosarcoma, Ewing sarcoma, Comprehensive Genomic Profiling, genomic alterations, genome-based therapies

## Abstract

The clinical heterogeneity of sarcomas, together with the complexity of their genomic landscape, has severely limited novel therapeutic opportunities. We report on our experience at the IRCCS Istituto Ortopedico Rizzoli using targeted gene sequencing for genome profiling with the intent of identifying potential actionable targets. We analyzed 22 advanced sarcoma patients. Genetic alterations in the NOTCH4, AR, BARD1, MUC16, and ROS1 genes, including missense, deletion, duplication, and delins, were the most frequent. Copy Number alterations affected the CDKN2A, CDKN2B, TP53, RHOA, MYC, CCND3, DDR2 genes. In four patients, longitudinal analysis of subsequent lesions highlighted an increase in mutations, like missense or splice variants in the PMS2, SMARCA4, ARID1A, AKT1, BMPR1A, and PTEN genes, suggesting tumor evolution. Our experience, aimed at refining clinical tumor profiling to identify potential therapies, highlighted the issue of the complexity introduced by mutational oncology and the primary role of the Molecular Tumor Board in the clinical management of advanced sarcoma patients.

## 1. Introduction

In recent years, the advent of Comprehensive Genomic Profiling (CGP) performed through Next Generation Sequencing (NGS) has transformed the accessibility of new therapies in the oncology field, enabling the simultaneous detection of a large number of somatic variants, both at diagnosis and during the medical or surgical treatment of the neoplastic lesions [1]. This is particularly true for oncological entities like breast, colorectal, and lung carcinomas, and cholangiocarcinoma [2,3], where there are well characterized, well known, and frequently recurrent variants and genetic aberrations that are common drivers of that type of tumors and that are directly related to therapeutic solutions [4].

For sarcomas, whether fusion-driven or not, the knowledge of the clinical genomic landscape and therefore the opportunity to link genomic variants to therapeutic opportunities is much less defined. Several whole-exome sequencing (WES) and whole-genome sequencing (WGS) analyses concerning sarcomas have been published [5,6,7,8,9,10,11,12], showing that these tumors present high intra- and intertumor heterogeneity. Sarcomas generally lack targetable fusion genes or driver mutations, present a low frequency of point mutations, and low values of tumor mutational burden (TMB) [8].

For these reasons, in sarcoma patients, targeted therapies or immunotherapies active in other tumor types failed to provide significant survival improvement [13,14] and standard of care treatments have remained unchanged over several decades. The prognosis of patients with metastatic/recurrent disease—not suitable for radical treatment—is generally poor. Currently available systemic therapies for advanced bone sarcomas are limited and provide only modest clinical benefit.

Many efforts are ongoing to explore the role of precision oncology in pediatric and adolescent patients, including sarcoma patients [15,16]. In 2024, the ESMO (European Society for Medical Oncology) recommendations for running tumor NGS in advanced cancer patients were updated to include sarcomas with reference to some entities of soft tissue sarcomas but not bone sarcomas [3]. The use of NGS in the clinical practice of sarcomas and bone sarcomas needs further refinement and is still at the exploratory level to evaluate if it can retrieve significant actionable targets and offer clinical benefit in advanced-stage disease.

Here, we present the evolving experience gained during 3 years (2022–2025) at the IRCCS Istituto Ortopedico Rizzoli, Bologna, Italy, in the context of a research project aimed at defining the optimal technical and procedural pathway to enable genomic profiling of patients with advanced-stage sarcoma. We report on our NGS experience in a small cohort of sarcoma patients relapsed or unresponsive to standard therapies with the aim of implementing genomic data into clinical management of rare and hard-to-treat tumors and potentially identifying targets for genomic-based therapies.

## 2. Materials and Methods

### 2.1. Patients and Samples

In the context of a research project directed to set up genomic profiling for advanced sarcoma patients, 22 patients with an indication by the oncologists to have CGP (185 genes analyzed) [17] performed were asked to sign a written informed consent form according to protocols approved by the institutional review board (CE AVEC660/2021/Sper/IOR; CE AVEC644/2023/Sper/IOR). Biological materials were stored in the Musculoskeletal Tumor Biobank of the IRCCS Rizzoli Orthopedic Institute.

Following consent, tumor patient samples and normal tissue samples (i.e., saliva using ORAGENE OG-5OO-DNA kit, Oragene, DNAgenotek, Ottawa, ON, Canada) were collected.

Accordingly, all tissue samples suitable for the molecular analysis were acquired from the Pathology Department after histopathological revision, assessment of tumor cell content, and overall evaluation of the quality of the specimen by an expert bone and soft tissue pathologist (MG). Extraction procedures were performed only on samples with 70% or more viable tumor cells. When multiple samples were available, the most recent one was selected for the first molecular analysis, while earlier disease-stage samples were subsequently analyzed, if available, to evaluate tumor evolution. When frozen tumor tissue was not available, Formalin-Fixed Paraffin-Embedded (FFPE) blocks were used instead.

### 2.2. Targeted Next Generation Sequencing

A PureLink genomic DNA mini-Kit (Invitrogen, Waltham, MA, USA) was used to isolate DNA from frozen tissue samples and saliva, whereas a Qiagen FFPE DNA mini kit (Qiagen, Hilden, Germany) was used for extracting genomic DNA from FFPE material according to the manufacturer’s instructions.

The concentration of nucleic acid was evaluated by the spectrophotometer NanoPhotometer N80-GO (Implen, Munich, Germany), and, before starting the library synthesis, the quality of the gDNA was assessed by qPCR to detect the number of correctly amplifiable genomes following the Archer PreSeq protocol (Archer PreSeq DNA QC Assay protocol).

The Archer’s VariantPlex Pan Solid Tumor kit panel (Archer, IDT, Coralville, IA, USA) used in this study is based on a targeted enrichment method called anchored multiplex PCR (AMP), which utilizes a protocol partially borrowed from RACE-PCR and represents an efficient way for specific amplification of DNA (either genomic or cDNA). This technique takes advantage of two different gene-specific primer sets (GSP1, GSP2) per target and allows for the amplification of DNA of low quality and/or from low amounts thereof [18].

The primer mixes of this panel allow for the simultaneous detection in 185 genes (Appendix A) of single and multiple nucleotide variants (SNVs/MNVs); copy number variations (CNVs), defined as loss or gain of entire copies of the gene; and deletion–insertions (delins), defined according to HGVS nomenclature recommendations [19] as a sequence change where, compared to a reference sequence, one or more nucleotides are replaced by one or more other nucleotides and which is not a substitution or inversion. In addition, this technology is able to perform an appraisal of Microsatellite Instability (MSI) over 114 genomic sites throughout the entire genome, and, lastly, on the basis of the release of the software used for the analyses, it can also give back the value of the Tumor Mutational Burden (TMB).

The libraries were synthesized following the manufacturer’s instructions. In accordance with the results of the qPCR number of amplifiable genomes detected, we started the libraries synthesis with a minimum of 50 ng of gDNA, up to a maximum of 200 ng of gDNA in order to be sure to always use, at least, more than double the 6500 amplifiable genomes per synthesis; therefore, the gDNA was fragmented and amplified using specific primer and enzyme blends provided in the kit. Libraries were then quantified using the KAPA library quantification kit (Roche, Basel, Switzerland) on an ABI 7900HT Sequence Detection System (Applied Biosystems, Waltham, MA, USA) and pooled to equimolar concentration. NGS was performed on a NextSeq-500 Platform (Illumina, San Diego, CA, USA) with MID Output 300 cycles flow cells and reagent cartridges. The results were analyzed using Archer Analysis software version 7.0 and 7.4.3.

### 2.3. Data Extraction and Plot Generation

All the data from the Archer Analysis Software was collected and organized in single research reports, one for each patient, following the requests given by oncologists. All indications for the detected variants were given, following the more recent guidelines from the Human Genome Variation Society (HGVS) Nomenclature [19], an internationally recognized standard for the description of DNA, RNA, and protein sequence variants, which is used to convey variants in clinical reports and to share variants in publications and databases.

SNVs and MNVs were filtered principally for having no Strand Bias and at least an Allele Fraction (AF) ≥ 0.40 (equal or greater than 40%, with an AF = 0.35 tolerance for genes with recognized therapeutic involvement).

More in detail, for calling somatic mutation, all the additional parameters used for filtering were: Alternative Observations, AO ≥ 5; Unique Alternative Observations, UAO ≥ 3; population AF ≤ 0.05 in gnomAD database; AF Outlier *p* Value ≤ 0.01; Sample Strand Bias = NO; Splice variants ≤ |3| bp from exon boundaries.

CNVs were considered if the CN was ≥2 for the amplifications and CN ≤ 0.5 for deletions. All the alterations were considered if they were calculated over more than 3 GSPs (Gene Specific Primers).

For the definition of MSI, it was considered the *p*-value of stability of each locus that should be lower than 0.05; therefore, a sample can be defined as MSI-Stable with less than 20 unstable loci, MSI-Intermediate with more than 20 unstable loci but less than 30 loci, and MSI-High when more than 30 unstable loci are discovered.

For the determination of TMB values, we computed all the variants spanning the covered areas of genome with the subsequent characteristics: Variant AF ≥ 0.1 and UAO ≥ 4; Coverage ≥ 70% bp with at least 200× coverage and Minimum Unique Fragments (on target) ≥ 410; TMB-Low < 5 Mut/Mbp; TMB-Intermediate in between 5 and 20 Mut/Mbp; TMB-High > 20 Mut/Mbp. We used CoMut, a Python library (version 3.0), to visualize genomic and phenotypic information [20]. All sequencing data that support the findings of this study are publicly available in the Sequence Read Archive (SRA) at BioProject and are accessible through the access number PRJNA1367486.

### 2.4. Droplet Digital PCR (ddPCR)

To validate the data of copy number alterations affecting the MYC gene in all 16 OS samples derived from 13 OS patients, we used Droplet Digital PCR (Bio-Rad Laboratories, Hercules, CA, USA) following the manufacturer’s protocol and using 4 ng of DNA. Taqman Copy number Assay (ID: hs01764918:cn) and TaqMan Copy number RNAseP human as reference assay (Applied Biosystems, #4403326) were used. PCR reaction mixtures (20 μL) were loaded into a DG8 cartridge (Bio-Rad Laboratories) with 70 μL of droplet generation oil through the Bio-Rad automated droplet generator system (Bio-Rad Laboratories). The generated droplets were then read using a QX200 droplet reader (Bio-Rad Laboratories) for the determination of total fluorescence. The data were analyzed with QuantaSoft version 1.7.4.0917 (Bio-Rad Laboratories). Based on the absolute quantification of the target molecules per reaction, the CNV was determined as the ratio between the target gene and the reference gene. Gain was defined as the detection of CNV > 3.

### 2.5. Identification of Potentially Actionable Alterations

Potentially actionable genetic alterations were identified using the Cancer Genome Interpreter (https://www.cancergenomeinterpreter.org/home (accessed along 2022–2025)) [21] and the OncoKB Database (https://www.oncokb.org (accessed along 2022–2025)) [22,23] tools, and categorized following the scale provided by the European Society for Medical Oncology (ESMO), which, in 2018, published the ESMO Scale of Clinical Actionability for molecular Targets (ESCAT) [24]. In brief, the ESCAT score is a clinical benefit-centered classification system and includes six levels and sub-levels with decreasing evidence of efficacy: (IA–IB–IC) Ready for routine use, alteration-drug match is associated with improved outcome in clinical trials; (IIA–IIB) Investigational, alteration-drug match is associated with antitumor activity but magnitude of benefit is unknown; (IIIA–IIIB) Hypothetical target, alteration-drug match suspected to improve outcome based on clinical trial data in other tumor types or with similar molecular alteration; (IV) preclinical evidence of actionability; (V) alteration-drug match is associated with objective response but without clinical meaningful benefit; (X) lack of evidence for actionability. According to ESMO guidelines [3], the Italian AIOM (Associazione Italiana di Oncologia Medica) (https://www.aiom.it/wp-content/uploads/2020/11/2020_RaccTumorBoardMolecolare.pdf (accessed on 14 November 2025)), and the ACC (Alleanza Contro il Cancro) (https://www.alleanzacontroilcancro.it/wp-content/uploads/2021/03/Linee-guida.pdf (accessed on 14 November 2025)) guidelines, a level of evidence at the ESCAT I/II level is required for the appropriate use of a drug.

## 3. Results

### 3.1. Clinical–Pathological Data

In this study, we analyzed a total of 22 patients over 3 years. In Table 1, we summarize the clinical–pathological characteristics of the patients, who were in the pediatric (0–14) and adolescent–young adult (15–39) ages, with a prevalent proportion of males compared to females. Tumors were classified according to the current WHO classification guidelines [25]. Globally, the most common histological subtype was osteosarcoma (OS), all conventional high-grade OS, followed by Ewing sarcoma (EWS), all carrying EWS::FLI1 translocation (type 1 in 1 case, type 2 in 2 cases, and not determined in 2 cases), and other rare tumors. Of note, in one CIC::DUX4 translocated sarcoma (CDS#2) and in three OS (OS#2, OS#5, OS#6) patients, two different tissue samples, taken in different stages of the progression of the disease, were available. Additionally, for nine patients, the corresponding saliva was made available, allowing for the investigation of germline DNA (Table 1). Seven out of twenty-two patients (32%) had metastatic disease at diagnosis, fourteen died of the disease (64%), one patient was lost at follow-up, and seven are alive with disease.

### 3.2. Targeted Sequencing Detected Relevant Genomic Alterations

For the first group of 13 patients, a total of 17 samples were analyzed, thanks to the availability of samples from subsequent lesions for 4 patients. No germline-matched samples were available. The comprehensive representation of all the alterations identified is shown in Figure 1. Collectively, the most frequent genetic variants were observed on the NOTCH4 (71%), AR and BARD1 (59%), MUC16 and ROS1 (53%) genes. Focusing on SNVs, missense variants represented the vast majority of the detected sequence alterations. They were followed by deletion–insertion (delins) alterations and other types of alterations.

Considering each tumor histotype, in OS, 5 out of 6 (80%) male patients presented sequence alterations (deletion and duplications) on the AR gene. Several alterations were detected in the NOTCH4 gene, showing deletion and delins in 5 out of 7 (71%) OS patients. Missense mutations were detected in the following genes: SMARCA4, in 6 out of 7 OS patients (86%); ARID1A, in 5 out of 7 OS patients (71%); PMS2, in 4 out of 7 OS patients (57%); BARD1, in 3 out of 7 OS patients (43%); ATR and POLD1 were both altered in 2 out of 7 OS patients (29%). ROS1 showed 2 delins and 1 missense mutation in 43% of OS patients. Splice-site mutations were found in other genes: PDGFRA (2 splice and 1 missense mutation out of 7 OS, globally 43%); AKT1 (3 splice mutations out of 7 OS patients, 43%); PTEN and SMAD4 (1 splice mutation each out of 7 OS patients, 14%). TP53 alterations, namely a copy number loss with CN = 0.19 in OS#1 and two missense variants (R273H in OS#2 A, and V216M in both OS#5 A and OS#5 B, two metastatic samples deriving from the same patient), were found only in OS patients (3 out of 7 patients, 43%), and not in other tumor histotypes. Regarding CNV, 4 out of 7 patients (57%) presented gains or losses on at least one gene. Patient OS#1 carried a loss of TP53 with a concomitant gain in FLCN, RHOA, and MYC genes. Patient OS#2 displayed a combined gain in CDK4 and CCND3 genes, and this CNV profile was present in both samples, primary tumor and recurrence, from the same patient. Patient OS#3 showed multiple gains on MUC16, AKT1, MSH2, and EPCAM with a loss on the two adjacent genes CDKN2A and CDKN2B. Patient OS#4 harbored a single gain on the DDR2 gene. Finally, all OS samples were defined as MSI-Stable.

In EWS, missense and delins alterations in NOTCH4, BARD1, MUC16, and PMS2 occurred in 75% of the patients (3 out of 4 cases) (Figure 1). Missense mutations in BMPR1A were observed in 50% of the patients. ROS1 was altered (1 splice-site mutation and 1 delins) in 50% of EWS patients, while PTEN showed a splice-site mutation in 1 out of 4 EWS patients. BRCA1 and BRCA2 showed 1 missense and 1 nonsense mutation, respectively, in 2 EWS patients. CNV analysis detected CN alterations in 1 out of 4 patients (CDKN2A loss in EWS#4). All EWS samples were characterized by an MSI-Stable profile.

For the 2 CIC::DUX4 translocated sarcoma patients, a total of 3 samples were available (Figure 1); two of them were derived from the same patient (CDS#2 B, primary tumor, and CDS#2 A, lung metastasis). In all 3 samples, the MET and MUC16 genes showed missense mutations. Sequence alterations were also detected in NOTCH4, AR, ROS1, BRCA1, POLD1, KDR, KMT2C, KMT2D, ABL1, FGF19, and TERT. PTEN showed a splice-site mutation in 1 patient. No CNVs were identified, and all samples were reported as MSI-Stable.

### 3.3. Longitudinal Analyses of Subsequent Samples from the Same Patient Allowed for Evaluation of Tumor Evolution

For 4 patients, we had multiple longitudinal samples available, taken in different stages of disease progression. For 1 CDS patient, we had a tissue sample from the primary tumor post-chemotherapy (CDS#2 B) and one from a metastasis (CDS#2 A). For the 3 OS patients: OS#2 A was a recurrence while OS#2 B was a biopsy of the primary tumor pre-chemotherapy; OS#5 A and OS#5 B were two subsequent lung metastases of the same patient with only one year of difference in between the two (OS#5 A, 2022 and OS#5 B, 2021); OS#6 A was a metastasis while OS#6 B was a biopsy of the primary tumor pre-chemotherapy (Figure 1). For each patient, when comparing the latest histology to the previous one, there was an increase in the total number of mutations in metastasis/recurrence samples versus primary tumor (CDS#2 A vs. CDS#2 B; OS#2 A vs. OS#2 B; OS#6 A vs. OS#6 B) (Figure 1). Accordingly, two lung metastases derived from the same patient (OS#5 A and OS#5 B) displayed a more similar mutational profile instead (Figure 1). The progressive enrichment in mutations in the longitudinal samples highlighted the occurrence of tumor evolution and the frequent involvement of selected families of genes. Among these, TP53 mutations, observed only in OS samples, showed a progressive increase in allelic fraction. In OS#2, the missense variant R273H was detected in the OS#2 B sample (biopsy), under the cut-off limit value (AF = 31.9%). Subsequently, it underwent an important gain in AF, ending with more than doubling the AF in the post-chemotherapy sample. Similarly, the V216M variant reported in OS#5 (two subsequent metastases) presented an AF of 62.4% in the first sample (2021, OS#5 B) and increased to 82.1% in the succeeding sample (2022, OS#5 A).

A group of genes associated with chromatin remodeling, the mismatch repair mechanism during DNA replication, and epigenetic modulation of gene expression (PMS2, SMARCA4, ARID1A, ARID1B) underwent missense mutations or deletions in subsequent samples of both CDS and OS (Figure 1), suggesting a progressive dysregulation of gene transcription and DNA repair. Of note, for the PMS2 gene (PMS1 Homolog 2, Mismatch Repair System Component) that we found mutated in both EWS and OS cases (75% and 40%, respectively), the four different types of missense variants that we found (K435E, T491S, K541E, and T597S) were all located between the mismatch repair domain and the C-terminal dimerization domain of the protein. Despite their neutrality score by bioinformatic tools of protein stability prediction (https://cancervar.wglab.org/ (accessed on 14 November 2025)), the K435E variant was present in 4 samples over the 8 samples harboring a PMS2 gene variant. Of note, this variant is known to be related to Lynch Syndrome 4 or hereditary nonpolyposis colorectal cancer type 4.

Splice-site mutations for AKT1 were found in 3 out of 4 subsequent samples for both CDS and OS, suggesting the dysregulation of signal transduction (Figure 1).

Splice-site mutations also occurred for PTEN in subsequent samples of CDS#2 and OS#6, suggesting alterations in oncosuppressor genes (Figure 1).

Overall, when multiple samples from the same patient could be analyzed, we were able to trace evolutionary processes along different samples, such as the accumulation of variants in different time-point samples and the evolution of the principal SNV/driver SNV across multiple samples with the progressively higher Allele Fraction value in samples coming from later stages of disease progression (Figure 1).

### 3.4. Matched Normal Samples Can Be Used to Accurately Identify Tumor-Specific Somatic Mutations

The further implementation of saliva analysis allowed for the subtraction of the preexisting germline variants. Therefore, only the variants proper of the somatic tumor sample were identified. The number of alterations discovered in the last series of cases was found, for this reason, to be lower compared to those previously retrieved in the different set of patients without matched normal sample analyses (Figure 2). The SEF sample showed the highest number of variants; the EWS sample, which did not show any alterations, was not reported in the figure. One of the most frequently altered genes was TP53, with 4 out of 6 OS (67%) samples and the SCOS sample showing missense mutations. Missense mutations were found in another tumor suppressor gene, SUFU, in 1 OS tumor and in the SCOS and SEF samples. In OS#8, the BRCA1 variant was reported, even if present in the germline sample, because the Allele Fraction of the variant in the somatic sample was drifting toward homozygosis. MSH3 and ARID1A genes showed one nucleotide duplication each, and a frameshift mutation was observed on ATRX.

Alterations in CNV were identified in 5 out of 6 (83%) OS samples and in the SEF sample. Loss of CDKN2A and CDKN2B was identified in one OS case (OS#9) and in the SEF sample as a coupled gene loss, due to the high genomic proximity of the two genes. In one case (OS#12), the loss encompassed only the CDKN2B gene. Somatic loss of RB was found only in one case (OS#13).

OS#12 presented the highest number of gains, including LZTR1, TRAF7, ERBB3, CDK4, and CCND3 (Figure 2). CCNE1 gain was seen in two OS samples (OS#8 and OS#10). Unexpectedly, MYC gain, frequently reported in advanced OS patients [12], was not detected in any of the 6 OS patients analyzed in this setting, and also considering the 7 OS patients of the previous group, its frequency was very low (1/13, corresponding to 8%).

The low frequency of MYC gain in our advanced sarcoma patients prompted us to validate the results by ddPCR. Data on copy number value from ddPCR for the MYC gene in all our 16 OS samples confirmed the results obtained from targeted next-generation sequencing, which showed a gain only in the OS#1 sample (Figure 1 and Figure 3 for comparison).

All samples were defined as MSI-stable except for one with an intermediate level of microsatellite instability (OS#13). We identified low TMB in seven cases, and intermediate TMB in one case (OS#9). Only the EWS sample did not match the criteria for minimum sequence coverage for the computation of TMB; therefore, the data was not available.

Similarly to the data reported in other studies [5], in our small cohort of patients, no correlation was found between the number of genetic alterations and the clinical outcome of the patients, but any correlation analysis was beyond the aims of our study. All the detected mutations, along with the ClinVar assessments of pathogenicity, according to the nomenclature from HGVSp and HGVSc, and their allele frequencies are listed in Appendix A.

### 3.5. Potential Actionable Targets

For all the cases analyzed, except one (EWS#5), potentially actionable alterations were reported to the clinical oncologists within 15 days from the request, in line with the time required by the AIOM and ACC guidelines. After an online analysis and comparison using the Cancer Genome Interpreter [21] and OncoKB Database [22,23], we identified a selection of drugs that were suitable for each genetic variants reported. However, when the gene alterations were classified according to the mandatory ESCAT Scale of Actionability [24] our actionable targets were always in the lowest part of the scale, never higher than III-A, which corresponds to the level of “Hypothetical target” with no sufficient clinical evidence and indications to support the treatment in the patient. Table 2 shows only the tumor-specific gene alterations identified for each case and the suggested drugs. Of note, as in other studies [16], we detected variants recognized to confer resistance to treatment with targeted drugs, underlining the utility of genomic profiling in predicting potential drug resistance.

## 4. Discussion

Personalized medicine needs the analysis of the genetic profile of the tumor to detect specific mutations or molecular alterations. This information can be used to identify relevant therapeutic targets within tumor cells, enabling the targeted use of drugs that may be most effective against cells harboring those specific variants. The aim of molecular profiling is to provide clinical oncologists with a choice of targeted treatments best-suited to the molecular profile of a patient’s tumor.

In recent years, several genomic studies have investigated the pathophysiology and genetics of sarcomas. For bone and soft tissue sarcomas, the use of WGS and/or targeted panels led to the identification of genetic heterogeneity with many chromosomal abnormalities, gene mutations, and down- and upregulated genes, including candidate driver genes [5,6,7,8,9,10,11,12].

In our study, we identified the highest frequency of mutation in the AR and NOTCH4 genes. AR mutations have been extensively studied in prostate cancer; their role in osteosarcoma and other sarcomas is less defined. Given the high frequency of AR alterations in OS lung metastasis and the observation that male patients with OS have a higher incidence of metastases compared to females and appear less sensitive to chemotherapy, the AR pathway has been proposed as a valuable therapeutic target against metastatic growth [26].

Molecules of the Notch signaling family are overexpressed in most clinical OS samples, correlating positively with recurrence, metastasis, and poor prognosis. Alterations in these genes modulate the sensitivity of OS to chemotherapy [27,28] and can undergo epigenetic regulation [29]. In our series, mutations (delins and deletion) in NOTCH4 were frequent findings in OS patients, occurring in 71% of OS cases.

In 70% of OS patients, we found mutations in the SMARCA4 and ARID1A genes. Both genes are components of SWI/SNF complexes, which are associated with chromatin remodeling, making genes accessible to transcriptional factors [30]. ARID1A is frequently mutated across human cancers, including sarcomas. ARID1A binds to DNA double-strand breaks (DSB) and to the DNA damage checkpoint kinase ATR. ARID1A mutations, therefore, hamper the DNA damage checkpoint activity [31]. Interestingly, ARID1A-mutated tumors were found to display high sensitivity to PARP inhibitors [32]. However, the significance of ARID1A in OS is far less explored.

Mutations in the BRCA1-associated RING domain 1 (BARD1) gene were found in both OS (40%) and Ewing samples (75%). The BARD1 protein, along with its heterodimeric partner BRCA1, plays an important role in cellular response to DNA damage. BARD1 is an obligatory binding partner of BRCA1/2 in the homology-directed DNA repair (HDR) complex when triggered for DNA damage repair [31,32,33].

Another gene involved in the mismatch repair mechanism during DNA replication is PMS2, for which we found mutations both in EWS and OS cases (75% and 60%, respectively). Among the four different types of missense variants found (K435E, T491S, K541E, and T597S), the K435E variant, related to the hereditary nonpolyposis colorectal cancer type 4 [34], was present in 4 out of 8 samples. In a study of variant analysis by whole-genome or whole-exome sequencing of tumor samples from 175 EWS patients, other types of missense pathogenic variants of the PMS2 gene (c.137G > T; p.S46I) were also identified. Considering that in EWS, the EWS::FLI1 fusion gene impairs HDR by sequestering BRCA1, the addition of PMS2 gene mutations probably further compromises HDR in EWS cells [35]. Collectively, these results identify the DNA repair pathways as strongly involved in malignancy and chemoresistance and indicate these alterations as highly relevant targets in sarcomas. Accordingly, a study on the genomic signature of homologous recombination deficiency estimated an occurrence of HDR deficiency in about 27% of OS samples [36]. In this context, it is strongly recommended the future integration of new CGP kits able to give an appraisal of the Genomic Instability Score (GIS), through the recording of all the variants linked to HDR proficiency, in order to evaluate the efficiency of the whole process of DNA repair in sarcoma samples and to consider the use of PARP inhibitors [37] in the clinical management of selected cases. Of note, a multi-feature genomic instability score has shown direct correlation with survival in soft-tissue and bone sarcoma patients [38], and HDR deficiency has been identified as a distinctive signature in sarcomas [39] and in osteosarcomas [40]. Despite the unsatisfactory results obtained till now in clinical trials employing PARP inhibitors in bone and soft tissue sarcomas [41,42], genomic profiling results seem to repurpose the evaluation of these strategies in eligible patients using selected types of PARP inhibitors and drug combinations.

Data obtained in larger cohorts reported a rate of TP53 alterations in 30–70% of OS samples [5,12,40]. In our study, TP53 alterations were found only in OS samples (7/13 cases, 54%) and in the unique sample of SCOS. Two missense variants, R273H and V216M, showed an enrichment of the Allelic Fraction in subsequent samples of the same patient, biopsy, and tumor recurrence for OS#2, and two subsequent lung metastases for OS#5. Both variants are defined as highly pathogenic and are associated with Li Fraumeni syndrome [43].

Longitudinal analyses of subsequent samples derived from progressive disease stages of the same patient allowed for the appraisal of tumor evolution that mainly involved the acquisition of mutations in TP53 and in genes playing a role in chromatin remodeling, DNA repair, and epigenetic modulation. Overall, the increasing number of variants and the progressively higher Allele Fraction values of driver SNVs observed over different time-point samples highlighted the need for the NGS analysis of tissue samples highly representative of the latest clinical situation of each patient. To this end, perhaps the utility of further biopsies of recurrences and metastases, as proposed in other studies [15,16], should be taken into account in relation to the patient’s clinical conditions and the individual harm–benefit ratio.

Considering copy number alterations, CDKN2A and CDKN2B were the most frequently deleted genes in our samples, showing coupled deletion, and this can be easily explained by their central role in regulating the cell cycle and by their topological proximity inside the short arm of chromosome 9 [44]. Gains affected a slightly higher number of different genes belonging to cell cycle regulation (AKT1, CCND3, CDK4, DDR2, EPCAM, FLCN, RHOA, CCNE1, and MYC) and DNA damage repair (MSH2), highlighting how these pathways play a fundamental role in the biology of sarcomas.

The role of the MYC oncogene has been extensively studied in osteosarcoma, and MYC amplification has been identified as a negative prognostic marker for survival [45]. Unexpectedly, in our small OS series, including patients with very aggressive disease, we found only one MYC amplification (8%), which was in the lower range of the data reported in the literature, which are in the order of 10–60% [5,12,40]. The validation of MYC copy number value by ddPCR confirmed the results obtained by NGS.

Loss of the RB1 oncosuppressor gene is a frequent finding in OS with a 10–60% incidence of alterations [5,12]. RB1 alterations occurred in our series, although with a frequency corresponding to the lower range recorded in the literature. Actually, we detected deletion of RB1 in one OS tumor as a specific acquired tumor somatic mutation corresponding to an 8% frequency of RB alterations in our small cohort of OS patients. In addition, it was previously noticed that RB1 alterations tended to be mutually exclusive with CDK4 and CDKN2A alterations [5], and mutual exclusion also occurred in our samples. Collectively, the low percentage of MYC amplification and of RB1 loss in our OS samples could probably be due to the limited number of patients analyzed here compared to other larger studies.

When the genomic analysis of saliva samples was implemented, the possibility of subtracting germline variants allowed for a simplification of the analysis and greater accuracy in the detection of specific tumor variants. The results obtained in this latter cohort of patients demonstrated that the variants detected were possible targets for specific drugs or had prognostic value (for example, for TP53 variants).

Finally, in our small cohort of patients, we did not find any sample with a high Microsatellite Instability profile, and all the samples had an overall low TMB value, in line with the data of the MAPPYACT study and the work of Gounder et al. [8], which, using a huge dataset, found that MSI-High sarcomas accounted only for 0.3% (18 over 6206) of the cases. The degree of TMB can predict a successful response to immunotherapy [46]; therefore, the low levels of TMB in sarcomas discouraged the recommendation of immunotherapeutic approaches.

Patient stratification in sarcomas is currently based mainly on histologic grade, tumor size, and anatomic site. The addition of genomic profiling could provide further prognostic information and potentially identify active targeted drugs in advanced sarcoma patients or define patient eligibility for future clinical trials evaluating targeted therapies [40].

In our setting, potential actionable alterations were identified in 95% of the patients. However, the ESCAT scores of actionability [24,47] were always in the lower part of the scale, never higher than level III. The genomic alterations detected in our study, while representing potential molecular targets, currently do not have sufficient clinical evidence to support the therapeutic use of targeted drugs in clinical practice. In sarcomas, the genetic landscape and the connection between variants and possible therapeutic solutions are far from being well assessed because all the therapeutic strategies associated with gene variants are almost all linked to tumors other than sarcomas.

In our series, CDK4/6 inhibitors were frequently suggested. CDK4/6 inhibitors in sarcomas exhibited multifaceted results. Frequent genomic aberrations of CDK4/6 in rhabdomyosarcoma, osteosarcoma, and other pediatric tumors were documented by means of the target actionability review (TAR) methodology to assess data on CDK4/6 as a therapeutic target in pediatric solid tumors. Despite the need for further preclinical and clinical investigations on the different oncological entities, this study recognized the relevance of CDK4/6 inhibition in Ewing sarcoma and, to a lesser extent, in rhabdomyosarcoma and other pediatric solid tumors [48]. However, a recent clinical trial on chemotherapy-refractory pediatric solid tumors, including osteosarcomas and rhabdomyosarcomas selected for harboring genomic alterations in the cyclinD-CDK4/6 and CCND3 pathway, concluded that pathway alteration alone was insufficient to generate a response to palbociclib (selective inhibitor of CDK4 and CDK6) monotherapy; therefore, underlining the need to further evaluate combined therapies [49].

A systematic review of the studies reporting on clinical outcomes in advanced patients treated according to molecular profiling highlighted an overall response rate (ORR) ranging from 0% to 67% [50]. A significantly improved rate of progression-free survival in patients treated following molecular indications, compared with those receiving conventional therapy, was obtained in controlled clinical trials [51,52], highlighting the need for additional prospective randomized controlled trials in selected tumor types.

In the European prospective multicenter precision medicine trial for advanced pediatric and adolescent cancer patients, the MAPPYACT study, which included 290 sarcoma patients, the outcome of patients treated with matched therapies was proportional to the level of evidence recommendations. For patients with “ready for routine use alterations”, the ORR was 38%. Lower-level evidence recommendations, “investigational” and “hypothetical”, resulted in a 14% and 10% ORR, respectively. Globally, the level of response was similar to that achieved in targeted drug trials, but superior when compared to chemotherapy trials [16]. Clinical actionability of genomic alteration in tumors has continuously increased in recent years [15,16,23]. However, sarcomas still suffer from the presence of many undruggable oncogenic drivers, and even for the potential druggable targets, the rarity of some sarcoma subtypes has hampered the evaluation of targeted drugs in randomized clinical trials, resulting in low levels of evidence in the ESCAT scale. Consensus guidelines for NGS in sarcoma diagnosis and treatment, which were developed recently in the context of the ESMO Rare Tumor & Sarcoma Annual Meetings [53], stated that, at present, routine clinical use of NGS for therapeutic purposes in sarcomas is not sufficiently supported by the available data due to the low number of actionable alterations with an adequate ESCAT score that can be expected for sarcomas, especially for rare subtypes. NGS should be employed in advanced sarcoma patients, informing them about its limitations, and using the most recent available sample. Data interpretation and its clinical use should be discussed by a multidisciplinary team of experts, possibly in reference centers for sarcomas [53].

## 5. Conclusions

Our experience has evolved deeply over time. The main elements that emerged from our study were as follows:The AMP chemistry and the 185-gene panel were adequate. Tissues from different sarcoma histotypes have different yields in terms of quantity and quality of the nucleic acids, and the AMP chemistry has proven to be robust and reliable in all situations, even if the DNA was extracted from FFPE tissues.Gene primer mixes able to detect almost every variant (SNVs or CNVs) connected to a therapeutic targeted solution can reduce the finding of Variant of Unknown Significance (VUS) and therefore simplify the process of interpretation of the molecular results. This is also true if we consider the inclusion of HDR proficiency evaluation with the definition of genomic instability score (GIS), which could be of great help in defining the eligibility of selected patients to PARP inhibitor treatment.A larger panel is highly recommended due to the fast pace of new drug–gene variant connections implemented every time the databases are updated to the latest releases and recommendations of the FDA, EMA, and other regulatory agencies. The possibility to offer genetic screening (both germinal and somatic) with larger panels (>400/500 genes) in an early stage of tumor progression will give the opportunity to deeply know the evolving traits of the tumor, and if there are familial syndromes linked to specific gene conditions, covering two analytical needs in one single step. The possibility of efficiently stratifying the patients will grant the appropriate enrolment in multiple trials aimed at assessing the efficacy of targeted drugs.We have been able to trace evolutionary processes along different samples, such as the accumulation of variants in different time points during tumor progression. This data highlights the importance of analyzing tumor samples representative of the latest disease stage of the patient and raises the issue of considering the utility of a second biopsy.

When considering advanced sarcoma patients, mutational oncology can offer new tools for the design of individual therapeutic programs. However, the governance of the complexity introduced by mutational oncology, the availability of targeted drugs, and the possibility of treating patients in an open clinical trial, along with the evaluation of cost-effectiveness and sustainability, requires collaboration among different specialists. To this end, many efforts are ongoing to develop institutional Molecular Tumor Boards for the proper use of molecular data and their integration with classical clinical–pathological information for the optimal management of advanced patients.

## Figures and Tables

**Figure 1 cancers-18-00139-f001:**
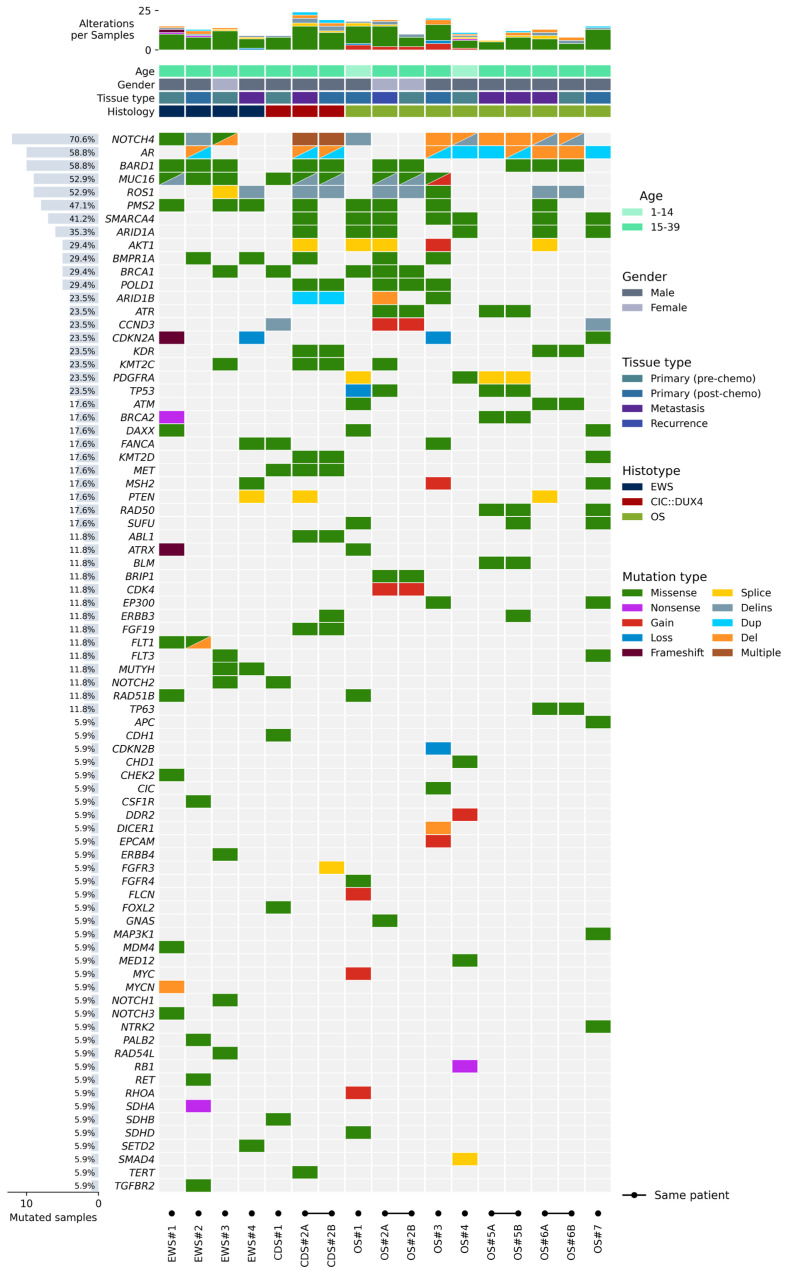
Frequency and type of genetic variants identified in 13 patients (corresponding to 17 samples, thanks to the presence of longitudinal histologies for 4 patients) by VariantPlex Pan Solid Tumor Kit. The number of alterations per sample and clinical–pathological features are depicted in the upper part of the figure, and genes are listed on the left. “Gain” and “Loss” refer to CNVs, i.e., the acquisition or the deletion of complete copies of the gene. Delins—deletion–insertion, Del—deletion, and Dup—duplication, all refer instead to changes involving a limited number of nucleotides and not entire copies of the gene.

**Figure 2 cancers-18-00139-f002:**
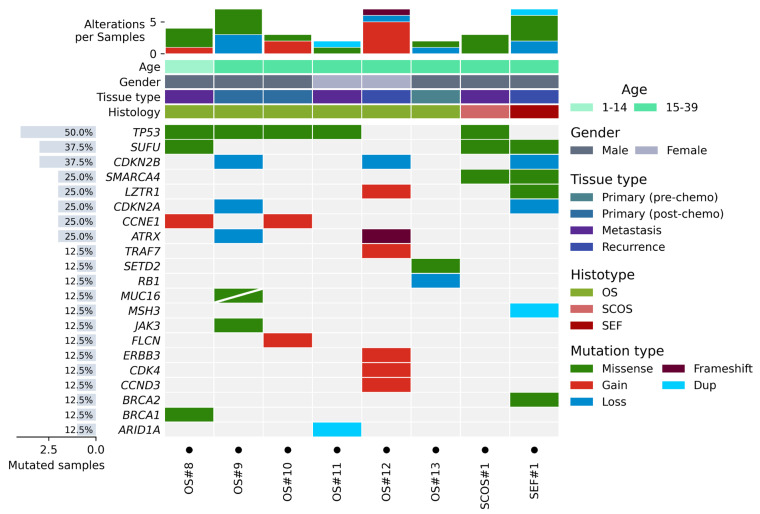
Frequency and type of genetic variants identified in 8 patients after the subtraction of the germline variants by the VariantPlex Pan Solid Tumor Kit. The number of alterations per sample and clinical–pathological features are depicted in the upper part of the figure, and genes are listed on the left. Sample EWS#5 did not show any alteration and therefore was not reported. “Gain” and “Loss” refer to CNVs, i.e., the acquisition or the deletion of complete copies of the gene. Delins—deletion–insertion, Del—deletion, and Dup—duplication, all refer instead to changes involving a limited number of nucleotides and not entire copies of the gene.

**Figure 3 cancers-18-00139-f003:**
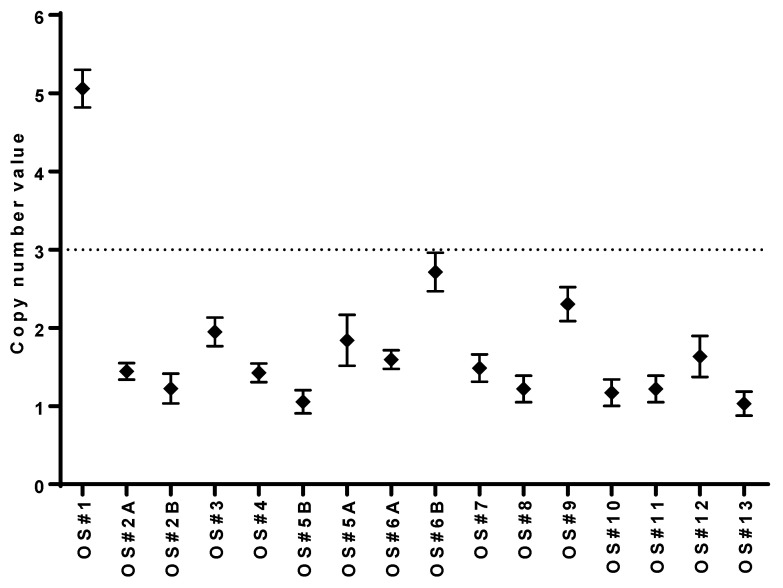
Validation of MYC gain by ddPCR. Each point represents a CN measurement from a single ddPCR experiment. Error bars indicate Poisson 95% confidence intervals for each copy number determination. Gain was defined as the detection of CNV > 3.

**Table 1 cancers-18-00139-t001:** Clinical–pathological features of the 22 patients analyzed in the study.

Patient and Sample Features	185 Genes Pan Solid Tumor Kit (Tumor)	185 Genes Pan Solid Tumor Kit (Tumor and Saliva)	Total (%)
**Age**			
0–14 years	2	1	3 (14%)
15–39 years	11	8	19 (86%)
**Gender**			
Female	3	3	6 (27%)
Male	10	6	16 (73%)
**Histology**			
Osteosarcoma (all high grade conventional OS)	7	6	13 (59%)
Ewing sarcoma	4	1	5 (23%)
CIC::DUX4 sarcoma	2		2 (9%)
Other sarcomas *		2	2 (9%)
**Tissue type**			
Primary (pre-chemo)	6	2	8 (36%)
Primary (post-chemo)	5	2	7 (32%)
Metastasis	2	3	5 (23%)
Recurrences		2	2 (9%)

* 1 sclerosing epithelioid fibrosarcoma (SEF), 1 small cell osteosarcoma (SCOS).

**Table 2 cancers-18-00139-t002:** Drugs identified using the Cancer Genome Interpreter and OncoKB databases for the potentially actionable tumor-specific molecular alterations. When multiple histologies were available, only the drug indications for the sample corresponding to the latest disease stage of the patient were reported.

Patient	Actionable Molecular Alterations	Drugs	ESCAT Score ^#^
**OS#8**	CCNE1:amp *	CDK2 inhibitors	IV-A

	TP53 (D281H) *	MDM2 inhibitors ^; Abemaciclib (CDK4/CDK6 inhibitor) ^; Cisplatin (Chemotherapy) ^; Doxorubicin (Anthracycline antitumor antibiotic); Gemcitabine; Mitomycin C; WEE1 inhibitors; AZD6738 (ATR inhibitor); Decitabine; Pramlintide (Amylin analog);	IV-A

	TP53 (D281H) ^§^	HSP90 inhibitors	IV-A
**OS#9**	CDKN2A:del *	CDK4/6 inhibitors; Ilorasertib (AURKA-VEGF inhibitor)	IV-A

	CDKN2B:del *	CDK4/6 inhibitors	IV-A

	TP53 (C275Y) *	MDM2 inhibitors ^; Abemaciclib (CDK4/CDK6 inhibitor) ^; Cisplatin (Chemotherapy) ^; Doxorubicin; Gemcitabine; Mitomycin C;WEE1 inhibitors; AZD6738 (ATR inhibitors); Decitabine;Pramlintide (Amylin analog).	IV-A

	JAK3 (T8M) ^§^	JAK inhibitors	X

	TP53 (C275Y) ^§^	HSP90 inhibitors	IV-A
**OS#10**	CCNE1:amp *	CDK2 inhibitors; Lunresertib AND/OR Camonsertib;	IIIA

	TP53 (D281H) *	MDM2 inhibitors ^; Abemaciclib (CDK4/CDK6 inhibitor) ^; Cisplatin (Chemotherapy) ^; Doxorubicin; Gemcitabine; Mitomycin C;WEE1 inhibitors; AZD6738 (ATR inhibitor); Decitabine; Pramlintide (Amylin analog)	IV-A

	TP53 (D281H) ^§^	HSP90 inhibitors	IV-A
**OS#11**	ARID1A(G84GGGGAGS) *	ATR, EZH2, PARP inhibitors	X

	TP53 (R282W) *	MDM2 inhibitors ^; Abemaciclib (CDK4/CDK6 inhibitor) ^; Cisplatin (Chemotherapy) ^;Doxorubicin; Gemcitabine; Mitomycin C; WEE1 inhibitors; AZD6738 (ATR inhibitors); Decitabine; Pramlintide (Amylin analog)	IV-A

	TP53 (R282H) ^§^	HSP90 inhibitors	IV-A
**OS#12**	CCND3:amp *	CDK4/6 inhibitors	X

	CDK4:amp *	LEE011, Abemaciclib + Palbociclib (CDK4/6 inhibitors); Palbociclib (CDK4/6 inhibitor) ^	III-A

	CDKN2B:del *	CDK4/6 inhibitors	IV-A

	ERBB3:amp *	EGFR mAb inhibitors ^	X
**OS#13**	RB1:DEL *	HDAC inhibitors; MDM2/MDMX inhibitors; Cisplatin;	IV-A

	SETD2 (D1166Y) *	WEE1 inhibitors	X
**SCOS#1**	TP53 (E258K) *	Cisplatin (Chemotherapy) ^; Abemaciclib (CDK4/CDK6 inhibitor) ^; MDM2 inhibitors ^; Doxorubicin (Anthracycline antitumor antibiotic); Gemcitabine; Mitomycin C; WEE1 inhibitors; AZD6738 (ATR inhibitor); Decitabine; Pramlintide (Amylin analog);	IV-A

	SMARCA4 (A321P) ^§^	EZH2 inhibitors	X

	TP53 (E258K) ^§^	HSP90 inhibitors	IV-A
**SEF#1**	CDKN2B:DEL *	CDK4/6 inhibitors	III-A

	CDKN2A:DEL *	CDK4/6 inhibitors; Ilorasertib (AURKA-VEGF inhibitor)	III-A

	BRCA2 (R2034C) ^§^	Olaparib; Rucaparib; Talazoparib; Niraparib (PARP inhibitors); Olaparib (PARP inhibitor) + Bevacizumab (VEGF mAb inhibitor); Talazoparib (PARP inhibitor); Enzalutamide (AR inhibitor); PD1 Ab inhibitors; Platinum Agent (Chemotherapy); Veliparib + Cisplatin (PARP inhibitor + Chemotherapy)	X

	MSH3 (A61APAAP) ^§^	DNA-PKc inhibitors	X

	SMARCA4 (A314P) ^§^	EZH2 inhibitors	X

* Indicate that the match between the sample alteration and the alteration for which the drug has been tested is complete. ^§^ Indicate that the match between the sample alteration and the alteration for which the drug has been tested encompasses other exonic variants for the same gene or has been approved for a different type of cancer. ^ Indicate that the mutation for which the drug is indicated confers resistance to that drug. ^#^ (The Cancer Genome Interpreter (https://www.cancergenomeinterpreter.org/home) accessed along 2024 and 2025).

## Data Availability

Sequencing data have been deposited in the publicly available database Sequence Read Archive (SRA) at BioProject and the accession number is as follows: PRJNA1367486.

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
