# Peer review of "Genomic Profiling of Highly Aggressive Musculoskeletal Sarcomas Identifies Potential Therapeutic Targets: A Single-Center Experience"

_cancers, 2025, doi:10.3390/cancers18010139_

Round 1

Reviewer 1 Report

Comments and Suggestions for Authors

The manuscript presents a solid and clinically relevant contribution to the genomic characterization of aggressive sarcomas. The integration of longitudinal sampling and germline–somatic comparison is particularly valuable. To further strengthen the work, the authors could elaborate on how their findings might guide future clinical trial designs or patient stratification strategies. Additionally, a deeper discussion on the functional impact of recurrent alterations—especially in DNA repair genes—would enhance translational relevance. Clarifying how the proposed larger panels and GIS assessment might be implemented in routine workflows would also be beneficial. Overall, the study is well executed and highly informative.

Author Response

The manuscript presents a solid and clinically relevant contribution to the genomic characterization of
aggressive sarcomas. The integration of longitudinal sampling and germline-somatic comparison is
particularly valuable.
1) To further strengthen the work, the authors could elaborate on how their findings might guide future
clinical trial designs or patient stratification strategies.
We added a comment on patient stratification strategies in the Discussion, on page 17, lines 8-11.
2) Additionally, a deeper discussion on the functional impact of recurrent alterations-especially in DNA
repair genes-would enhance translational relevance. Clarifying how the proposed larger panels and GIS
assessment might be implemented in routine workflows would also be beneficial.
The issue has been addressed in the Discussion, on pages 15-16, and in the Conclusion, on page 18, lines 24-
30.
Overall, the study is well executed and highly informative.

Reviewer 2 Report

Comments and Suggestions for Authors

This is a good study. The sample size is small, so it is hard to obtain generalized conclusions. The study provides no new information, although the methodology and reporting are adequate and appropriate. This manuscript can provide guidance to future researchers in conducting experiments. Clinicians can derive benefit by using the analytical methods used in the manuscript to help find drugs for potential targets. 

It would be really nice if the sample size can be increased to at least 50 patients before we publish to help increase the credibility of the findings. 

Author Response

This is a good study. The sample size is small, so it is hard to obtain generalized conclusions. The study provides no new information, although the methodology and reporting are adequate and appropriate. This manuscript can provide guidance to future researchers in conducting experiments. Clinicians can derive benefit by using the analytical methods used in the manuscript to help find drugs for potential targets. 

It would be really nice if the sample size can be increased to at least 50 patients before we publish to help increase the credibility of the findings. 

-We would like to thank the reviewer for the helpful hint. Unfortunately, increasing the number of patients is
not possible at the moment. Surely a larger cohort would have been more informative, but the aim of this
article was to report the experience of two research projects that are now concluded with their small cohort
enrolment. We will take advantage of this observation in the future for new research project.

Reviewer 3 Report

Comments and Suggestions for Authors

The authors use the term “delins” for deletions/insertions. I am more familiar with the term being “indels” for insertion/deletion. I had never heard the term delin used previously. In looking this term up, some define it as complex indel with insertion and deletion occurring at the same genomic site. The authors may wish to use indel to avoid confusion.

It might be useful to define at what point an indel becomes a CNV.

The key for Figure 1 is very confusing. Multiple pairs mutation type have very similar colors (e.g. nonsense and gain, frameshift and dup, splice and del). The key has a mutation type “multiple” but multiple types of mutation in one gene in one patient appear to be denoted by coloring the box half one and half the other.  Finally, the mutation types are poorly defined. Each abbreviation should be spelled out in the legend or text. I am also unclear as to the meaning of each abbreviation. For example, what are “gain” and “loss”? It would seem they mean insertion and deletion, but there is also a type “delin” and type “del” (for deletion?) and frameshift could be due to an insertion or deletion. While figure 2 is less complicated, the same issues are present.

Author Response

1) The authors use the term "delins" for deletions/insertions. I am more familiar with the term being "indels" for insertion/deletion. I had never heard the term delin used previously. In looking this term up, some define it as complex indel with insertion and deletion occurring at the same genomic site. The authors may wish to use indel to avoid confusion.

  • We used the term "delins" according to the most recent adjournments of HGVS Nomenclature. At the link: https://hgvs-nomenclature.org/stable/recommendations/DNA/delins/
  • We have better clarified the meaning in the Materials and Methods, on page 3, lines 38-47, and on page 4, lines 15-18.

2) It might be useful to define at what point an indel becomes a CNV.

  • For what concerns delins we can state that it deals about substitutions of a reduced number of nucleotides (generally not more than 10 and surely not the sequence of an entire exon). In our cases it deals about 3-5 nucleotides changed from the reference sequence. The CNV instead is the loss or gain of a complete gene sequence.
  • We have better specified the meaning of CNV in the Materials and Methods, on page 3, lines 38-47.

3) The key for Figure 1 is very confusing. Multiple pairs mutation type have very similar colors (e.g. nonsense and gain, frameshift and dup, splice and del). The key has a mutation type "multiple" but multiple types of mutation in one gene in one patient appear to be denoted by coloring the box half one and half the other. Finally, the mutation types are poorly defined. Each abbreviation should be spelled out in the legend or text. I am also unclear as to the meaning of each abbreviation. For example, what are "gain" and "loss"? It would seem they mean insertion and deletion, but there is also a type "delin" and type "del" (for deletion?) and frameshift could be due to an insertion or deletion. While figure 2 is less complicated, the same issues are present.

  • We amended figures 1 and 2 caption and the figure colours in order to reduce the possibilities of confusion and to clarify the information given.
  • The other terms "Delins", "Del", "Dup" were described in figure legends 1 and 2.

Reviewer 4 Report

Comments and Suggestions for Authors

In this manuscript, the authors profile 22 advanced sarcoma patients on a gene panel, 9 with matching germline samples. Since the majority of these did not have matching germline samples, the mutation calls can be difficult to assess and derive interpretations.

Major comments

  1. Since this is a small cohort of rare cancers, please list the mutations, along with the ClinVar assessments of pathogenicity if available, using the nomenclature from HGVS, and the allele frequencies of the mutations for Figures 1 and 2.

  1. Figure 1: It is strange that there are no losses of RB1 in these samples, especially for osteosarcoma. Also the frequency of TP53 deletions seems low for osteosarcoma. Can the authors comment on this? Since there is no matching germline sample, many methods should be used for calling somatic mutations including excluding variants with population allele frequencies >0.01 using databases such as GNOMAD, using an in silico-derived matched normal that combines several matched normals together, using a germline caller (found in GATK or platypus).

Minor comments

  1. Figure 2 is the calls cleaned up with the germline mutations. Are there supposed to be 8 or 9 patients here since 9 germline samples were described in the first paragraph of the Results.
  2. Please indicate the SRA accession number for the dataset.

Author Response

Since this is a small cohort of rare cancers, please list the mutations, along with the ClinVar assessments of pathogenicity if available, using the nomenclature from HGVs, and the allele frequencies of the mutations for Figures 1 and 2.

-All the variants have been reported in an excel file (supplementary table 2) with the complete description using HGVSp. and HGVSc. conventions. A sentence has been inserted in Results, paragraph 3.4, page 12, lines 10-13.

Figure 1: It is strange that there are no losses of RB1 in these samples, especially for osteosarcoma. Also the frequency of TP53 deletions seems low for osteosarcoma. Can the authors comment on this? Since there is no matching germline sample, many methods should be used for calling somatic mutations including excluding variants with population allele frequencies >0.01 using databases such as GNOMAD, using an in silico-derived matched normal that combines several matched normals together, using a germline caller (found in GATK or platypus).

-The issue of the unexpected low frequency of alterations in specific genes has been discussed, see page 16, lines 36-45.

-In the Materials and Methods section, paragraph 2.3 we better indicated the filtering using gnomAD database included in the Variant Calling (Archer Analysis Unlimited) and considering population AF below 0.05 (see page 4, lines 22-24). However, we will take advantage of this observation for the future studies strengthening the filtering criteria in order to be even more specific in the variant calling process.

Minor comments

Figure 2 is the calls cleaned up with the germline mutations. Are there supposed to be 8 or 9 patients here since 9 germline samples were described in the first paragraph of the Results.

-The reason for this apparent incongruence is explained in the figure 2 legend: "Sample EWS#5 did not show any alteration and therefore was not reported".

Please indicate the SRA accession number for the dataset.

-SRA accession number has been indicated in the Materials and Methods section, paragraph 2.3, page 4, lines 39-41.

Reviewer 5 Report

Comments and Suggestions for Authors

Review Summary

This manuscript investigates the genomic profiling of highly aggressive musculoskeletal sarcomas and identifies potential therapeutic targets. The study is well written, and the findings are of significant interest to practicing oncologists, surgeons, and pathologists. However, there are several points that require clarification and additional detail before the manuscript can be considered for publication.

Major Comments

  1. Pathology Expertise
    Please clarify in the Materials and Methods section whether the pathology diagnoses were rendered by an expert bone and soft tissue pathologist.
  2. Classification Standards
    Were the tumors classified according to the current WHO classification guidelines? If so, please specify and include the WHO Classification of Soft Tissue and Bone Tumors as a reference.
  3. Table 1 Clarification
    In Table 1, please indicate the osteosarcoma subtype (e.g., conventional high-grade or low-grade).
  4. Fusion NGS Profile
    What was the fusion NGS profile of the Ewing sarcomas? Providing this information would strengthen the manuscript.
  5. Correlation Analysis
    I recommend including additional details on the correlation between clinical outcomes and mutational profiles, as this would enhance the translational relevance of the findings.

Recommendation

Major Revisions
The manuscript has merit and addresses an important topic, but the above points should be addressed to improve clarity and completeness.

Author Response

Major Comments

  • Pathology Expertise Please clarify in the Materials and Methods section whether the pathology diagnoses were rendered by an expert bone and soft tissue pathologist.

This indication has been added in the Materials and Methods, paragraph 2.1, see page 3, lines 18-19.

  • Classification Standards Were the tumors classified according to the current WHO classification guidelines? If so, please specify and include the WHO Classification of Soft Tissue and Bone Tumors as a reference.

This indication and related reference have been added in the Results, paragraph 3.1, see page 5, lines 30-31.

  • Table 1 Clarification In Table 1, please indicate the osteosarcoma subtype (e.g., conventional high-grade or low-grade).

This information has been added in the Results, paragraph 3.1, see page 5, lines 31-33, and in table 1.

  • Fusion NGS Profile What was the fusion NGS profile of the Ewing sarcomas? Providing this information would strengthen the manuscript.

This detail has been added in the Results, paragraph 3.1, see page 5, lines 31-33.

  • Correlation Analysis I recommend including additional details on the correlation between clinical outcomes and mutational profiles, as this would enhance the translational relevance of the findings.

Clinical outcome of the patients has been inserted in the Results section paragraph 3.1, page 5, lines 37-39. Considering the low number of patients, we did not find any correlation between clinical outcome and number of mutations, and we inserted a statement and a reference in the Results, paragraph 3.4, page 12, lines 8-10.

Round 2

Reviewer 5 Report

Comments and Suggestions for Authors

Thank you for addressing the recommended changes in the manuscript. The revisions have significantly improved the content. No further changes are required.